# Ultrasensitive rapid cytokine sensors based on asymmetric geometry two-dimensional MoS$_2$ diodes

Thushani De Silva [1], Mirette Fawzy[2], Amirhossein Hasani[1], Hamidreza Ghanbari[1], Amin Abnavi[1], Abdelrahman Askar[1], Yue Ling [1], Mohammad Reza Mohammadzadeh[1], Fahmid Kabir [1], Ribwar Ahmadi[1], Miriam Rosin[3], Karen L. Kavanagh[2] & Michael M. Adachi [1] ✉

The elevation of cytokine levels in body fluids has been associated with numerous health conditions. The detection of these cytokine biomarkers at low concentrations may help clinicians diagnose diseases at an early stage. Here, we report an asymmetric geometry MoS$_2$ diode-based biosensor for rapid, label-free, highly sensitive, and specific detection of tumor necrosis factor-α (TNF-α), a proinflammatory cytokine. This sensor is functionalized with TNF-α binding aptamers to detect TNF-α at concentrations as low as 10 fM, well below the typical concentrations found in healthy blood. Interactions between aptamers and TNF-α at the sensor surface induce a change in surface energy that alters the current-voltage rectification behavior of the MoS$_2$ diode, which can be read out using a two-electrode configuration. The key advantages of this diode sensor are the simple fabrication process and electrical readout, and therefore, the potential to be applied in a rapid and easy-to-use, point-of-care, diagnostic tool.

Cytokines are small proteins that play an important role in regulating the inflammatory response. Found in biofluids such as blood, saliva, and sweat, they have gained interest as biomarkers for various health conditions and diseases[1–3]. An abnormal change in cytokine concentration is an indicator of uncontrolled inflammatory reactions that has been linked to Alzheimer's disease, cancers, pulmonary tuberculosis, autoimmune, and cardiovascular disease[1,4–7]. In addition, coronavirus 2019 (COVID-19) infection is accompanied by a release of an elevated level of pro-inflammatory cytokines such as interleukins (IL-1β and IL-6) and tumor necrosis factor-α (TNF-α), in an occurrence called a cytokine storm[8]. Studies have suggested that cytokine inhibitors are an effective treatment for improving COVID-19 survival[9]. Treatment of many diseases is most effective at an early stage[10]. Thus, the ability to monitor and detect early changes in inflammatory cytokine levels is of great interest to clinical diagnosis. Serum levels of TNF-α among healthy young and adult population is typically in the range of

200–300 fM[11]. In the case of children, the serum levels can be as low as 12 fM[12]. Hence, having a limit of detection (LOD) in the range of fM is important in early diagnostic applications.

The standard method for measuring cytokines is via an enzyme-linked immunosorbent assay (ELISA), a technique which is widely used in clinical laboratories and biomedical research[13]. Single molecule array (SIMOA), an ultrasensitive ELISA method, and mass spectroscopy can detect cytokines at concentrations in the fM range, sufficiently sensitive to monitor diseases in an individual[14]. However, these methods are time-consuming and expensive, limiting widespread use for diagnostic applications.

Biosensors are analytical devices that consist of a biorecognition element (the receptor) on a transducer, which transforms the interactions between the biorecognition element and the specific target into a measurable signal[15]. There are a number of different sensing mechanisms in biosensors, including optical, electrical, acoustic and

[1]School of Engineering Science, Simon Fraser University, Burnaby V5A 1S6 BC, Canada. [2]Department of Physics, Simon Fraser University, Burnaby V5A 1S6 BC, Canada. [3]Department of Biomedical Physiology and Kinesiology, Simon Fraser University, Burnaby V5A 1S6 BC, Canada. ✉e-mail: mmadachi@sfu.ca

electrochemical.[16,17]. For example, Ghosh S et al. reported detection of TNF-α using a quantum dot-based optical aptasensor with a LOD in the pM range[18]. A malaria biomarker employing an antibody-aptamer plasmonic biosensor reported a LOD of 18 fM[16].

Among the different types of biosensors, field-effect transistor (FET) based biosensors have attracted great attention by providing a label-free, rapid, low power consumption, portable, and highly sensitive sensing platform that can be easily integrated with other electronic components such as data analyzers and signal transducer applications[2]. However, FET biosensors, including graphene-based FETs (GFETs) and FETs based on transition metal dichalcogenides (TMDs), are generally used in a liquid-gated configuration with an Ag/AgCl electrode, which can hinder the integration and scaledown of the device[2]. The gate electric field, applied via a sample solution, can disturb the binding affinity between the charged cytokines and receptors, hence affecting the sensing stability[19]. Also, the continuous electrical stress in the liquid can lead to undesirable leakage current, that would generate false sensor response and electronically damage the sensor[20]. A serious issue related to TMD-based FET sensors is the hysteresis in the transfer characteristics arising due to gate-modulated charges trapped at the TMD/dielectric interfaces. This behavior makes the current measured across the drain and source, under a given gate voltage, highly dependent on the sweep range, sweep direction, sweep time and loading history of gate voltage biases, which leads to inconsistent sensor readings[20]. A solid-gated FET employing a dielectric layer could mitigate some of the issues present in the liquid-gate sensors. However, the solid-gate FETs, particularly devices with thick SiO₂ dielectric layer, typically require high operating gate voltages in the range of 40–50 V for GFETs and ~100 V for TMD-based FETs[20], which can be a human health hazard[2,21].

By employing a simple two-electrode diode sensor, issues associated with gating can be completely avoided. A number of recent publications have reported the diode rectification behavior arising due to an asymmetry in contact geometries for several 2D-materials including graphene, WSe₂,WS₂, In₂S₃, and GeAs[22–27], and also in a varying diameter Si nanowire (NW)[28]. An asymmetric contact geometry also offers a simple fabrication process requiring only one metallization layer (Cr/Au) and no doping process.

Here, we report the detection of the pro-inflammatory cytokine TNF-α, at concentrations as low as 10 fM using an asymmetric geometry diode, with a wide dynamic range of detection (5 orders of magnitude) from concentrations of 10 fM to 1 nM. We employ a well-known silane functionalization process and aptamer cytokine interaction to demonstrate the detection of cytokines using this device architecture. The device's rectification factor (RF), the ratio of current measured at reverse and forward bias voltages (−1 and +1 V), changes as a function of cytokine concentration, when added to the sensor surface in a phosphate-buffer saline solution (PBS). The ultra-sensitivity, we presume is due to the change in charge density at the surface of the TMDs, generated by cytokine-aptamer interactions. Our biosensor offers several key advantages compared to existing methods of cytokine measurement including a smaller sample volume, (3 μl of TNF-α diluted in PBS compared to the typical volume of 100 μl used for ELISA), rapid measurement without the need for incubation, minimal hands-on time, and no requirement for expensive optical equipment. Moreover, the diode sensor uses only two electrodes, which simplifies the fabrication process and readout electronics compared to field-effect-transistors (FETs), which require a third gate electrode. Due to its simple operation, and absence of complicated post-measurement analysis, the diode sensor requires minimal training and is therefore suitable for point-of-care diagnostic applications. Since the detection of TNF-α inherently depends upon the bioreceptor (aptamer) anchored to the sensing area, the proposed sensor application can be extended to detect other cytokines, proteins, or other biomarkers molecules by replacing the bioreceptor with one that specifically binds to the corresponding biomarker. Due to the flexibility and mechanical stability of 2D MoS₂, a flexible and wearable biosensor device for health monitoring applications is feasible. Considering these advantages, we anticipate that our cytokine biosensor has the potential for application as an easy-to-use, point-of-care biomarker testing tool.

## Results

### Aptamer-based cytokine diode sensor

An illustration of the cytokine measurement procedure using an asymmetric geometry MoS₂ diode is shown in Fig. 1. Note that the blood sample shown in Fig. 1a is a conceptual illustration of how TNF-α concentrations would be measured in a blood sample. The sensor consists of a 2D semiconductor, (2H-phase) multilayer MoS₂ crystal flake on top of thermally-oxidized SiO₂ (oxide thickness of 300 nm) contacted by two Cr/Au electrodes. Based on atomic force micrscopy

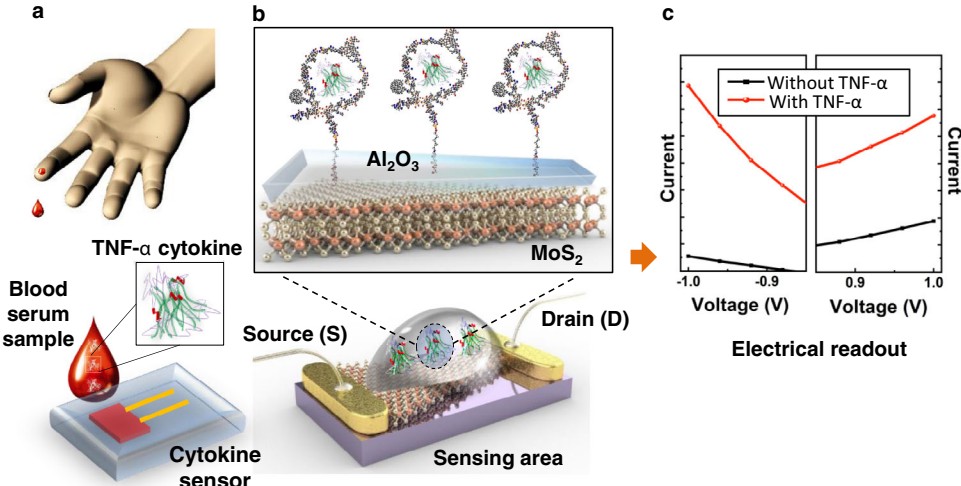

**Fig. 1 | Schematic illustration of the concept of the cytokine sensor operation. a** A small volume of blood serum is drop casted onto the sensing area. **b** The cytokine sensor consists of an asymmetric geometry MoS₂ crystal contacted by two metal electrodes. The inset shows a magnified diagram of the sensing area showing how TNF-α cytokines are bound to aptamer receptors on the oxide, forming G-quadruplex structures and bringing charged cytokines closer to the surface of the sensor. **c** The change in surface charge density induces a change in the electrical rectification behavior of the MoS₂ diode observed in a current–voltage (I–V) measurement.

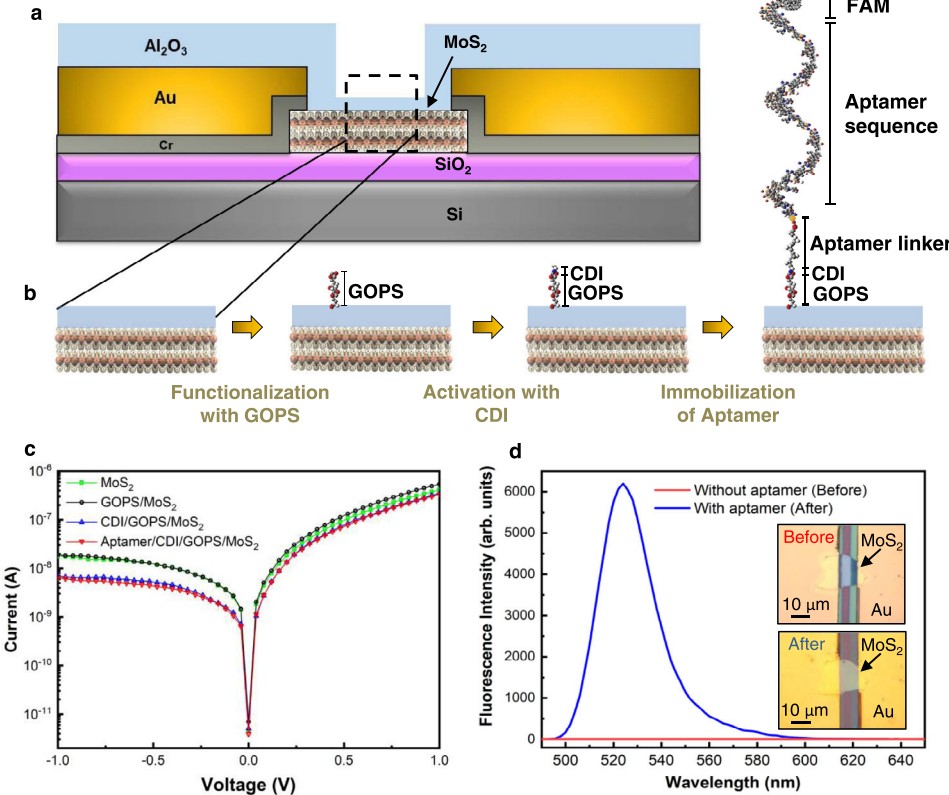

**Fig. 2 | Functionalization of the asymmetric geometry diode sensor. a** A cross section schematic diagram. **b** Sensor functionalization consisting of (glycidoxypropyl) trimethoxysilane (GOPS) and 1-1'carbonyldiimidazole (CDI) linkers that covalently couples the TNF-α binding aptamer to the Al₂O₃ coated sensor. **c** I–V curves of the sensor during different stages of the functionalization process. **d** Fluorescence measurement used to confirm that the aptamers, labeled with a fluorescent dye (FAM dye), were immobilized on the sensor surface. The insets show the micrographs of device sensing area before and after functionalization.

(AFM) measurements, the typical MoS$_2$ thicknesses were between 13 and 60 nm (Supplementary Fig. 1). Cr/Au contacts were fabricated with photolithography to form two electrical contacts across the flake, with an electrode spacing of 10 μm. The geometric asymmetry of the MoS$_2$ crystal, with two MoS$_2$-metal contacts of different cross-sectional lengths and areas, induces a diode rectification behavior. The MoS$_2$ flake is coated with a thin insulating Al$_2$O$_3$ layer, which is then functionalized with an aptamer that specifically binds to the targeted cytokine TNF-α. When a sample containing TNF-α (e.g., the blood serum in Fig. 1a) is drop cast onto the sensor, the TNF-α cytokines bind to the aptamer forming a G-quadruplex structure that leads to negatively-charged TNF-α moving closer to the sensor surface. Changes in surface charge density induces a change in the electrical rectification behavior in the asymmetric geometry MoS$_2$ diode.

The surface potential across the device area was measured using Kelvin probe force microscopy (KPFM). The surface potential maps were measured on the asymmetric geometry MoS$_2$ diode across the longer and shorter metal-semiconductor interfaces (Supplementary Fig. 2a, b, respectively). The surface potential barrier measured across the longer and shorter MoS$_2$-metal interfaces (along the line scan shown in Supplementary Fig. 2a, b) shows a difference in Schottky barrier contact which arises due to different contact area[23]. The appearance of the rectification behavior due to the flake asymmetry is further supported by data from other asymmetric and symmetric MoS$_2$ devices fabricated in a similar method as shown in Supplementary Fig. 3. No significant rectification was observed for symmetric flakes. The asymmetric barriers at the two MoS$_2$-metal interfaces give rise to the diode rectification behavior as shown in Fig. 1c. That is, the absolute value of current is asymmetric between −1 and +1 V in the initial device, before introducing TNF-α. The exposure of TNF-α to

the sensor surface induces a change in rectification behavior in the current–voltage curve, and the relative change in rectification behavior corresponds to the concentration of the TNF-α introduced.

A cross-section schematic of the asymmetric geometry diode sensor is shown in Fig. 2a. An Al$_2$O$_3$ insulating thin film layer deposited by atomic layer deposition (ALD) technique[15,29] performs the following roles: (1) facilitates the aptamer functionalization; and (2) electrically isolates the metal contacts from the buffer solution containing the cytokine analyte. The thickness of the Al$_2$O$_3$ is 5 nm above the MoS$_2$ crystal, which is the active sensing area between the metal electrodes, and 75 nm elsewhere, including on top of the electrodes, which provide electrical isolation.

## Functionalization of the device

The details of the functionalization process via silane chemistry using (glycidoxypropyl)trimethoxysilane (GOPS) and 1-1'carbonyldiimidazole (CDI) linkers are included in the Methods section. A step-by-step process flow of the sensor functionalization is illustrated in Fig. 2b. GOPS couples to the Al$_2$O$_3$ surface in an aqueous solution at a low pH environment[30]. The GOPS is activated by the attachment of a CDI linker[30] followed by coupling between the amine on the 5' end of the TNF-α-binding aptamer to the GOPS active sites. A DNA aptamer is used as the biomarker receptor since it binds specifically to the target analyte TNF-α,[31,32] and is potentially reusable[33]. To facilitate the coupling of the aptamer oligonucleotide on the sensing surface, the devices were immersed in a 10 μM aptamer in PBS solution.

The I–V characteristics from a device at different steps of the functionalization process are shown in Fig. 2c. The pristine device (coated with Al$_2$O$_3$) initially displayed a rectification factor, *RF* (log $|I_{-1V}|$ − log $|I_{+1V}|$), of -1.4. After the GOPS functionalization step, it

increased slightly to -1.5 and after the CDI functionalization step, it further increased to -1.7. The rectification was almost unaffected during the aptamer coupling to the activated GOPS step. The first step of the functionalization involves the hydroxyl groups at the $Al_2O_3$ surface reacting with the GOPS in which the epoxide ring on the GOPS opens to produce a diol[30]. These diols are activated by CDI which creates more amine-targeted binding sites[30]. These surface modifications induce a change in the surface potential which can cause the changes in the rectification observed in Fig. 2c. However, during the final step, the amine-modified aptamers in a PBS solution couple only to the CDI moieties and do not induce a strong change in the overall rectification of the device.

The aptamer oligonucleotides used in this study had a fluorescent dye (FAM) on the 3' end so that the successful coupling of aptamers to the sensor could be verified by fluorescence spectroscopy. Figure 2d shows the fluorescence spectrum measurements obtained with and without aptamer functionalization. The emission spectrum of the FAM (peak emission at 525 nm) was absent in the bare sample. The inset of Fig. 2d shows optical images before and after sensor functionalization.

## Electrical measurements for sensing TNF-α

An optical image of an asymmetric geometry diode sensor contacted by two probe tips is shown in Fig. 3a. The $Al_2O_3$ dielectric layer was removed at the end of the gold contact lines, over the contact pads (purple region), to facilitate an ohmic electrical connection between the probes and pads (Supplementary Fig. 5). Since the rectification of the $MoS_2$ Schottky diodes are sensitive to light, all the measurements were carried out in the dark[34,35].

The I–V responses obtained for a single device as a function of cytokine concentration is shown in Fig. 3b (log scale). As the concentration of the cytokine was increased, an increase in $RF$ was observed. A magnified view of the I–V curves at high voltages (linear scale) is shown in Fig. 3c.

Due to variations from device-to-device in $MoS_2$ flake geometries and thicknesses arising from the mechanical exfoliation process, and in the aptamer functionalization process (fluctuations in room temperature and humidity), a normalized rectification factor, $RF_N$, was used to compare different devices, defined as:

$$RF_N = \frac{RF - RF_{PBS}}{RF_{max} - RF_{PBS}} \qquad (1)$$

where $RF_{PBS}$ is the $RF$ with only PBS (0 fM TNF-α in Fig. 3b, c) and $RF_{max}$ is the maximum $RF$ for each individual device (typically observed at the maximum TNF-α concentration)[36]. The $RF_N$ as a function of TNF-α cytokine concentration (presented as $x$-axis data), is plotted in Fig. 3d. The data was fit with a classical Hill function[37]:

$$RF_N = \frac{1.05}{1 + 10^{(-3.08-x)0.26}} \qquad (2)$$

where 1.05 = top asymptote (A1) − bottom asymptote (A2), −3.08 = log of center of $x$-axis data (log x0), and 0.26 = hill slope (p). A reduced $\chi^2$ statistic of 0.0003 was obtained for the fitted curve.

Sensor specificity is achieved by employing an aptamer DNA sequence, referred to as VR11, which has been shown to have high specificity to TNF-α cytokine[31,32,36]. The specificity of our sensor was investigated by introducing two non-target inflammatory biomarker proteins, IL-6 and C-reactive protein, to the sensor using the same conditions as TNF-α. The $RF_N$ for three different non-target protein concentrations (0.01, 1, 100 nM) alongside TNF-α is shown in Fig. 3e. For the non-specific IL-6 cytokine, the $RF_N$ response was close to zero at all three concentrations. On the other hand, the non-target C-reactive protein showed a small $RF_N$ response at 0.01 nM and a noticeably higher response at higher concentrations (1 and 100 nM). Still, the

highest response observed for C-reactive protein was 3 times lower than the TNF-α cytokine response at the same concentration. A possible cause for the lower specificity seen in the C-reactive protein case could be due to its higher molecular weight, which may cause a considerable number of proteins to physically adsorb to the sensing area without being bound to the aptamer. A similar behavior has been reported by Fathi-Hafshejani P et al. for a $WSe_2$-based FET, functionalized with a continuous layer of a 2.7 nm long linker for the detection of SARS-CoV-2 virus, where a considerable response to pure ionic liquid used as a negative control was observed[38]. A method to improve specificity of a sensor is to use a surface passivation technique, such as ethanolamine, after the aptamer functionalization step to block non-specific binding at unreacted sites on the sensing area[30].

To further verify that the response observed in the sensor is due to the successful binding of the TNF-α cytokine to the aptamer, a negative control test was carried out on another asymmetric geometry diode sensor prepared using the same fabrication process and functionalized using GOPS and CDI linkers but without aptamers. The $RF_N$ response of the control device without aptamer functionalization and the sensor that was fully functionalized with aptamer, for different TNF-α concentrations, is shown in Fig. 3f. A stark contrast can be observed wherein the $RF_N$ of the aptamer-functionalized sensors changed as a function of TNF-α concentration while the control device response fluctuated around the zero level.

## Detection mechanism

TNF-α molecules, diluted in PBS buffer solution (1×) (pH ~7.4), are deemed to be negatively charged[31,36]. When a cytokine binds to a TNF-α specific aptamer, the aptamer folds, forming a stable and compact G-quadruplex, which causes the negatively charged cytokine, along with the electron-rich aptamer end to come closer to the $Al_2O_3$ surface as illustrated in Fig. 4a. As a result, there is an increase in negative charge on the $Al_2O_3$ surface inducing a change in the rectification factor in the $MoS_2$ sensor[31,36]. Since the aptamer changes its form upon binding to a target, the fluorescence intensity changes depend on the manner of the FAM dye modification[39,40]. In this case, the fluorescence intensity is expected to decrease after the cytokine interaction due to the aptamer folding. Hence, the decrease in the fluorescence intensity could be regarded as an indicator of the change in the aptamer structure into a compact from, bringing the charged cytokine closer to the surface as shown in Supplementary Fig. 6. Note that the purpose of the fluorescent FAM dye is to verify that aptamers are successfully bound to the sensor surface and confirm aptamer-cytokine interaction and is not needed in the measurement of TNF-α.

A liquid gating measurement, illustrated in Fig. 4b, was performed to support the proposed detection mechanism. Figure 4c shows the I–V response between the drain and source ($I_{DS}$) Au contacts, obtained for a diode sensor under increasing negative liquid gate voltage ($V_{GS}$) from 0 to −1 V, applied via a droplet of pure PBS solution to the sensing area. The I–V response shows a noticeable fluctuation in current between −0.75 and 0 V which we believe is due to the presence of PBS since the I–V response in air did not display such behavior (Supplementary Fig. 7a). It has been reported that the graphene-based FET biosensors are susceptible to disturbance occurring in the capacitance across the electrical double layer formed at the solution-graphene interface[2,41,42]. With the thin dielectric layer over the sensing area (5 nm), it is possible that our diode sensors undergo a similar disturbance during the liquid-gating which causes the current to fluctuate at low voltage values. Gate leakage current for each I–V response shown in Fig. 4c were measured to be negligible (Supplementary Fig. 7b), and the AFM height image over the 5 nm $Al_2O_3$ film in the sensing area shows that the oxide layer is continuous and free of pinholes (Supplementary Fig. 7c). The $RF_N$ increases with increasing negative $V_{GS}$ (Fig. 4d), which is similar to the response seen in sensor

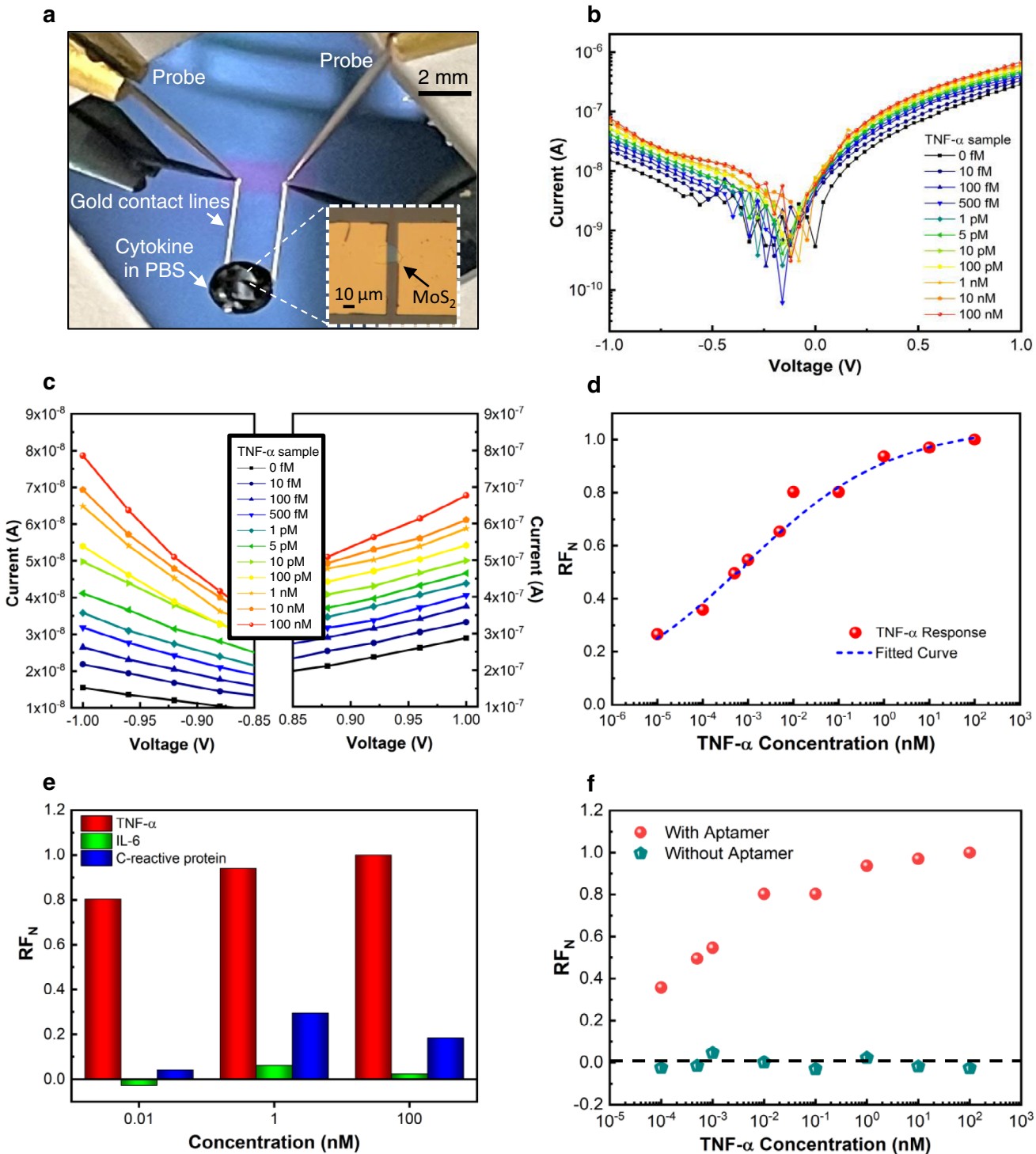

**Fig. 3 | Performance of the sensor. a** An optical image of the TNF-α measurement setup where the sensing material, MoS₂, is contacted by electrical probes. The inset shows an optical micrograph of the device sensing area while coated by a drop of TNF-α in phosphate-buffered saline (PBS). *I–V* response of the MoS₂ diode as a function of TNF-α cytokine concentration in **b** log scale, and **c** linear scale with a magnified view. **d** Normalized rectification factor, $RF_N$, as a function of TNF-α cytokine concentration. **e** Specificity of the detection of TNF-α against two other non-targeted proteins, IL-6 and C-reactive protein. **f** $RF_N$ as a function of TNF-α concentration for a device with and without aptamers.

output during the interaction between TNF-α to the aptamer measured in Fig. 3b–d. Hence, the trend seen in Fig. 4d is a strong indication that the binding of TNF-α to the aptamer prompts changes in the surface charges in the MoS₂ sensing layer, which induces a change in the diode rectification.

Another noticeable observation in the *I–V* response is the increase in the current level as a function of both increasing TNF-α

concentration (Fig. 3b) and increasing negative gate voltage (Fig. 4c). Conventionally, a negative gate voltage prompts a decrease in drain-source current in FET transistors using n-type materials, such as MoS₂ crystals grown by chemical vapor deposition[43]. Defects such as S vacancies are likely to be in abundance in such material, enhancing n-type behavior[44]. However, in mechanically exfoliated thin MoS₂ flakes, which have less S deficiency, an enhanced p-channel with more

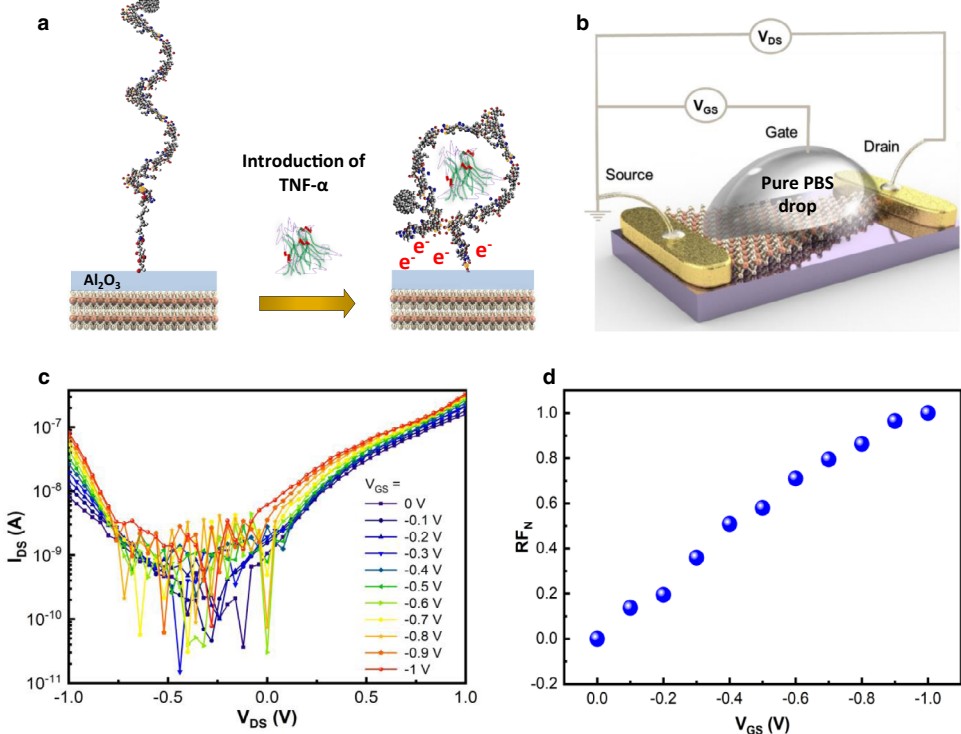

**Fig. 4 | Detection mechanism. a** Illustration of the folding of an aptamer, forming a compact G-quadruplex structure, due to the interaction of the TNF-α cytokine and aptamer, which brings negative charges closer to the $Al_2O_3$ thin film coated sensor surface. **b** Schematic diagram of a setup to support the proposed detection mechanism by investigating the effect of gate voltage on $I$–$V$ response of a liquid-gate, asymmetric-geometry, diode sensor. **c** Drain-source current ($I_{DS}$) versus drain-source voltage ($V_{DS}$) for different applied gate voltages ($V_{GS}$) ranging from 0 to −1 V with an increment of −0.1 V steps (the gate voltage was applied via a pure PBS drop as shown in Fig. 4b). **d** $RF_N$ versus $V_{GS}$ of the liquid gated sensor.

balanced ambipolar transport can be realized[45]. In fact, Ye T. J. et al. and Zhang Y et al. demonstrated the ambipolar operation in $MoS_2$ by the observation of an increase in hole current with increase in negative liquid gate voltage[45,46]. Similar to these observations, the increase in the current level with the cytokine concentrations/negative liquid gate voltage measured in cytokine diode sensor shows the existence of ambipolar transport in the thin $MoS_2$ flakes used in this study which was confirmed by the transfer curve shown in Supplementary Fig. 8.

## Discussion

Table 1 summarizes TNF-α biosensing techniques reported in the literature, including the sensing material, receptor type used, and the reported LOD. Electrochemical biosensors based on detecting biological analyte through redox reactions on electrodes[47] have been shown to achieve LODs in the fM range[48,49]. However, compared to electrical sensors such as FET-based biosensors, the specificity of the electrochemical biosensors is lower. The FET biosensors have displayed an enhanced specificity in detecting a targeted cytokine[31,36]. Even so, typical FET cytokine sensors that have employed graphene as the sensing material, have reported a relatively high LOD, in the pM range. The lack of a bandgap in graphene fundamentally limits its sensitivity and reduces the dynamic range of the biosensor. It has been shown that FETs using 2D $MoS_2$ as the channel material can be 70 times more sensitive than graphene FETs in biosensing applications[50]. $MoS_2$ FET-based biosensors used in the detection of various cytokine including TNF-α biomarkers have been reported with LODs in the range of 60–400 fM[31–54].

In this study, we demonstrated the rapid detection of TNF-α with a LOD of 10 fM, which is amongst the lowest concentrations reported using biosensor devices. We believe that the relatively low LOD was achieved due to the nature of the device structure consisting of a thin $Al_2O_3$ passivation layer (5 nm) over the sensing $MoS_2$ area, combined with a thicker passivation layer (75 nm) over the electrode areas which minimized leakage currents. A similar strategy was used for a $MoS_2$ FET sensor based on antibodies, employing a thick $SiO_2$ film to passivate the contacts and a thin $HfO_2$ layer on the $MoS_2$ channel, with a LOD of 60 fM[52]. Since the aptamers used in this work are smaller in size compared to antibodies, they offer improved transportation of the target towards the sensor's surface[55]. Particularly the aptamer sequence used in this study, VR11, can bring the charged TNF-α more closer to the sensor surface upon affinity binding[31]. Our initial device fabrication process contained a single thin passivation layer to cover the sensing area as well as the gold electrodes. However, device degradation occurred during measurement due to possible oxidation of the $MoS_2$ flakes (Supplementary Fig. 9). To mitigate this issue, the passivation was carried out by a two-step process where a thicker $Al_2O_3$ layer was used to passivate the gold electrodes and a thinner layer covered the sensing area to passivate the sensing material, $MoS_2$, and facilitate the aptamer functionalization.

The performance of the sensor is also affected by the initial rectification factor of the asymmetric diode before functionalization, which is generally higher for flakes with a triangular shape having a large asymmetry in metal-$MoS_2$ overlapping area between the two contacts. In this study, mechanically exfoliated flakes were used to demonstrate the biomolecule detection method. For commercial applications, a lithographic control over the flake geometry will be needed to make reproducible and reliable sensors for clinical usage with up-scalable potential. By employing a large-area compatible process to prepare thin films of $MoS_2$[56], the sensing area can be patterned into regular triangular shapes using lithography and etching to have better control over the flake geometry. With the advancement of large area 2D material synthesis, scale-up of diode sensors is possible. For example, large area graphene oxide films cut into asymmetric geometries have been employed to develop a nanofluidic pH sensor

**Table 1 | Comparison of biosensors for the detection of TNF-α, listing the technique, the sensing material, receptor type used, and the reported limit of detection (LOD)**

| Technique | Sensing material | Receptor type | LOD | Reference |
|---|---|---|---|---|
| | | | nM | |
| Reflectometric interference spectroscopy | Amino-acid decorated | Nano structured | 3.3 | [59] |
| QCM | Gold electrode | Antibody | 1.5 | [60] |
| Potentiometric sensor | PVA nano shell | Antibody | 0.8 | [61] |
| Electrochemical redox spectra | Gold electrode | Aptamer | 0.3 | [62] |
| Electrochemical redox spectra | Gold electrode | Aptamer | 0.3 | [63] |
| Photonic micro ring resonator | Gold electrode | Antibody | 0.3 | [64] |
| | | | pM | |
| PL spectroscopy | Quantum dot | Aptamer | 99 | [18] |
| Electrochemical | Gold electrode | Aptamer | 58.5 | [33] |
| FET | Graphene | Aptamer | 26 | [31] |
| FET | Graphene | Aptamer | 5 | [36] |
| FET | Graphene | Aptamer | 2.8 | [65] |
| Plasmon resonance (LSPR) | Gold nanorod | Antibody | 0.6 | [66] |
| Electrochemical | Indium tin oxide | Antibody | 0.6 | [67] |
| Electrochemical | Graphene oxides | Antibody | 0.3 | [68] |
| Electrochemical impedance | $TiO_2$ nanotube | Antibody | 0.3 | [69] |
| | | | fM | |
| FET | $MoS_2$ | Antibody | 60 | [52] |
| Electrochemical | Gold electrodes | Antibody | 58.5 | [70] |
| Photoelectrochemical | $TiO_2$ nanorod and ZnS nanoparticle | Antibody | 58.5 | [49] |
| Capacitance electrochemical | Silicon nitride | Antibody | 58.5 | [48] |
| Diode sensor | $MoS_2$ | Aptamer | 10 | This work |

based on ion-current rectification[25]. There have been advances in synthesizing wafer-scale 2D materials such as single crystal graphene on germanium substrates[57] and epitaxially grown $MoS_2$ on sapphire substrates, which can be employed to scale up the sensor fabrication process[58].

In summary, we present a method to detect TNF-α, at concentrations as low as 10 fM combined with a wide dynamic range of detection between 10 fM to 1 nM, based on an asymmetric geometry $MoS_2$ diode sensor. The TNF-α measurement was via an $I–V$ response curve obtained during a 2 min interaction between a diluted TNF-α solution and an aptamer-functionalized sensor surface. The sensor presented here has shown promising results for rapid detection of femtomolar concentrations of TNF-α using a simple two electrode design, making it suitable for easy-to-use and rapid point-of-care testing.

## Methods

### Materials and chemicals
Bulk 2-H phase single crystal $MoS_2$ was supplied from SPI Supplies. Glycidoxypropyltrimethoxysilane (GOPS), 1,1'-carbonyldiimidazole (CDI), and acetonitrile (ACN) were purchased from Sigma-Aldrich. Hydrochloric acid (HCl) was purchased from Fisher Scientific. Molecular biology grade water/ nuclease-free (N-free water) and 1× phosphate-buffered saline (PBS) was purchased from Lonza. TNF-α-specific aptamer (VR11) with the fluorescent tag (sequence /5AmMC6/ TGG ATG GCG CAG TCG GCG ACA A/36-FAM/) was synthesized and purified by Integrated DNA Technologies. Recombinant Human TNF-α Protein, Recombinant Human C-Reactive Protein, and Recombinant Human IL-6 Protein were purchased from Bio-Techne.

### Fabrication of the $MoS_2$ diode
The fabrication steps are illustrated in Supplementary Fig. 4, where the first step was to exfoliate $MoS_2$ flakes onto a Si/SiO₂ substrate. Before exfoliation, the Si/SiO₂ substrates were cleaned by sonication in acetone for 10 min, 2-propanol for 10 min and distilled water (DI water) for 10 min. Flakes were prepared by mechanical exfoliation and selected for device fabrication based on their shape (with geometric asymmetry) observed under an optical microscope. Metal electrodes were patterned across the flake by deliberately introducing an asymmetry in the metal-semiconductor interface (area and length) using photolithography. Next, a 10 nm of Cr and 50 nm of Au were deposited for electrical contacts. The initial gap across the electrodes was kept between 10 and 20 μm. Afterwards, for the passivation purpose, a 70 nm thick $Al_2O_3$ layer was deposited via atomic layer deposition (ALD) at 250 °C. A thicker passivation layer effectively minimized the leakage current between electrodes[52]. A narrow strip was patterned (using positive photoresist) in the middle of the electrode gap (over the $MoS_2$ flake) keeping a margin of ~2 μm at each side. Then the $Al_2O_3$ strip was completely etched in BOE (until the flake was exposed). Finally, a second layer of $Al_2O_3$ was deposited with a thickness of 5 nm.

### Functionalization and activation of the sensing area
To functionalize the sensing area, we followed a method reported by Potyrailo R. A. et al.[30] who used a SiO₂ surface. First, the pristine diode sensors were submerged in a 10 % aqueous solution of GOPS where the pH was maintained at 3.5 using HCl. After degassing for 10 min (using N₂), the reaction was allowed to proceed at 90 °C for 4 h with occasional shaking[30]. The devices were then washed with acetone and 2-propanol and then placed to dry in an oven at 60 °C overnight. The surface was activated by submerging the devices in saturated CDI-acetonitrile solution and shaking (1.5 h at 20 °C). Finally, the devices were rinsed with N-free water.

### Immobilization of aptamers
As received amine-modified ssDNA aptamers were reconstituted at a concentration of 100 μM using diluted PBS and aliquoted to 50 μl volumes, which were then stored at −20 °C as per the

manufacturer's instructions. The activated devices were submerged in a 10 μM aptamer solution (prepared using the stored samples) for 24 h. Afterwards, the unreacted aptamers were rinsed using N-free water, then dried with $N_2$, and stored at −20 °C until measurement[30].

## Cytokine sample preparation

As received TNF-α cytokine samples were reconstituted to a concentration of 25 μg/ml and aliquoted to 20 μl volumes which were then stored at −20 °C. A similar method was followed in reconstituting and aliquoting IL-6 cytokine. C-reactive protein was reconstituted at a concentration of 6 μg/ml and aliquoted to 500 μl volumes for storing at −20 °C. For the experiments, different cytokine concentrations ranging from 10 fM to 100 nM, were prepared using these stored samples by diluting in PBS.

## Characterization and sensing measurements

Atomic force microscopy (AFM, Asylum MFP3D) was employed to measure the thickness of the $MoS_2$ flakes (Supplementary Fig. 1). KPFM (Bruker AFM System) measurements were carried out to map the surface potential difference (Supplementary Fig. 2). During the KPFM measurements, both sides were grounded. Fluorescent measurements were carried out by employing a fluorescence imaging spectrometer (HORIBA iHR 320) with an excitation laser wavelength at 485 nm. The data was acquired using a 10× objective lens. The acquisition time was set at 10 s with 3 accumulations. The range of the spectrum was selected to be from 490 to 650 nm with a step size of 2 nm. Electrical measurements were performed using a Keithley 4200-SCS semiconductor characterization system connected to a probe station. The Si substrate was placed on an electrically insulating stage in the probe station and was electrically isolated. All $I–V$ measurements were conducted in the dark at room temperature in atmospheric pressure with a scan speed of 40 mV/s.

Cytokine in PBS solution was drop cast onto the sensing area (Fig. 3a) at a volume adequate to cover the sensing area (2–3 μl) and left for 2 min to react with the sensor surface after which an $I–V$ response measurement was taken over an applied voltage of −1 to 1 V. The cytokine in PBS solution drop was removed using an air blower and the next cytokine concentration in PBS solution was immediately dropped onto the sensing area.

## Statistics and reproducibility

Due to the nature of the experiment in this work, each sensor could only be used one time (for one series of concentrations). Once a sensor reaches the saturation point and most of the receptors are bound to the target, it cannot be used for another round of testing. The method of preparing the sensing material, $MoS_2$, was mechanical exfoliation, which produces a random distribution of $MoS_2$ flakes having different shapes and thicknesses. Since each device was different from one another, the standard deviation of the sensor output was not calculated. Measurements were taken from distinct devices. We have shown that the normalized rectification factor as a function of cytokine concentration has the same trend in multiple devices, thereby demonstrating the reproducibility of the sensor output. This information is included in the Supplementary Fig. S10.

## Reporting summary

Further information on research design is available in the Nature Portfolio Reporting Summary linked to this article.

## Data availability

Source data is available for Figs. 2c, d, 3b, d–f, and 4c, d and Supplementary Figs. 1a, b, 2a, b, 3a–f, 6, 7a, b, 8, 10a–e in the associated source data file. Source data are provided with this paper.

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

## Acknowledgements
This work was supported by grants from the Government of Canada's New Frontiers in Research Fund (NFRF) – Exploration [NFRFE-2018-00214] (M.M.A., M.R., and K.L.K.), Natural Science and Engineering Research Council (NSERC) [RGPIN-2017-05810] (M.M.A.), Canada Foundation for Innovation (CFI) [Project No. 38331] (M.M.A.), British Columbia Knowledge Development Fund (BCKDF) (M.M.A.), Western Economic Diversification Canada (WD) [Project No. 000015280] (M.M.A.), and Simon Fraser University. The authors thank B. Kim and A. Szigeti for maintenance of the SFU Engineering Science cleanroom facility and Dr. D. Leznoff and W. Zhou for access to Raman microscopy. The authors acknowledge CMC Microsystems and 4D LABS shared facilities that facilitated this research.

## Author contributions
T.D. and M.F. contributed equally to this work. M.R., K.L.K., and M.M.A. conceptualized the project and provided project leadership. T.D. designed the sensor. T.D. and M.F. performed all fabrication and experimental tasks under the supervision of M.R., K.L.K., and M.M.A. A.H. designed the schematics diagrams and participated in data analysis. T.D., M.F., H.G., and Y.L. functionalized the sensors. A. Abnavi, A. Askar, and R.A. assisted with sensor design, atomic layer deposition, and KPFM measurements. M.M. and F.K. assisted with device fabrication. All authors participated in data analysis and discussion. The manuscript was written through contributions of all authors. All authors have given approval to the final version of the manuscript.

## Competing interests
T.D., M.F., M.R., K.L.K., and M.M.A. are inventors on a provisional patent application on the topic of this work filed by Simon Fraser University. The remaining authors declare no competing interests.
