## [Peer Review File · Nature Communications]

Ultrasensitive rapid cytokine sensors based on asymmetric geometry two-dimensional MoS₂ diodesReviewers' comments:

Reviewer #1 (Remarks to the Author):

The authors present an electronic biosensor consisting of an asymmetric MoS₂ channel, resulting in a nonlinear diode-like transfer characteristic. The basic biosensing mechanism still stems from the traditional antigen-antibody binding mechanisms. The presented sensing results for cytokine molecules exhibit a good detection limit down to 10 fM. The whole device fabrication is based on the exfoliation method followed with regular lithographic steps. The authors have presented very detailed description for the device fabrication and characterization. However, the overall quality of this manuscript is not suitable for Nat. Commo. First, MoS₂-based or other TMDC-based nanoelectronics biosensors have been extensively investigated. For example, this previous work ACS Sens. 2017, 2, 2, 274 reported 1 fM LOD for IL-1 beta quantification. Most cytokine molecules exhibit a relatively large affinity with their corresponding antibodies. Therefore, it is not very exciting nowadays to see a fM-level LOD under the presented testing configuration. Second, the presented device fabrication process lacks a upscalable potential. I would like to recommend this manuscript transferred to other journals focusing on alternative biosensing techniques.

Here are more comments:

1. The MoS₂ channel thickness is reported to be in the range of 13-60 nm. However, the optical micrograph inset in Fig. 3a shows a MoS₂ flake that looks significantly thicker than this range. Please use AFM to double check the thickness values.
2. The authors claim that the rectification effect of the MoS₂ device is attributed to the different contact areas at two MoS₂/electrode interfaces. However, previous works have shown that many fabrication defects or contaminations can result in very strong rectification effects. How can these factors be ruled out?
3. It is still not clear why the diode-like I-V response characteristics result in the improvement of the biosensing sensitivity. A device physics model is needed to support the presented arguments.

Reviewer #2 (Remarks to the Author):

The paper reports detection of an inflammation biomarker using an asymmetric MoS₂ diode. Key noteworthy results are- small sample volume of 3 μ l, rapid measurement without the need for incubation and LOD of 10 fM.

However several questions related to the detection mechanism along with the reproducibility and variability of the results need to be addressed.

(1) The authors claim that the diode current increases due to increased hole current. To show enhanced ambipolar transport, please show the I_d vs V_G for negative V_D , negative V_G and positive V_D , positive V_G combinations before and after the various functionalization steps. One explanation for Fig. 4c data is an increase in negative charge in MoS₂ which opposes the claim of negative charge on the bio-molecules. Another explanation is increased gate leakage (current through Al₂O₃). Based on this, please show gate leakage data for these I-V measurements.

(2) Please provide AFM data of Al₂O₃ surface after the etch showing lack of pin-holes and a continuous film.

(3) Why is the data so noisy between -0.75 V and 0 V in Fig. 4c?

(4) Were the devices washed before and after each of the functionalization steps? This would ensure covalent binding as reported and no physical adsorption.

(5) What is max RF in a given device for a particular concentration? is it $(\log|\max I_+(+1V)| - \log|\max I_-(-1V)|)$ for a single maximum value in a single graph?

(6) Please provide data from various devices to show reproducibility and variability with error bars.

(7) Is there a way to confirm the increase in negative charge on the bio-molecules after cytokine binding through say KPFM or other measurement techniques? Also for the bending of the molecules?

Response to Reviewer #1

Reviewer comment: The authors present an electronic biosensor consisting of an asymmetric MoS₂ channel, resulting in a nonlinear diode-like transfer characteristic. The basic biosensing mechanism still stems from the traditional antigen-antibody binding mechanisms. The presented sensing results for cytokine molecules exhibit a good detection limit down to 10fM. The whole device fabrication is based on the exfoliation method followed with regular lithographic steps. The authors have presented very detailed description for the device fabrication and characterization. However, the overall quality of this manuscript is not suitable for Nat. Commo. First, MoS₂-based or other TMDC-based nanoelectronics biosensors have been extensively investigated. For example, this previous work ACS Sens. 2017, 2, 2, 274 reported 1 fM LOD for IL-1 beta quantification. Most cytokine molecules exhibit a relatively large affinity with their corresponding antibodies. Therefore, it is not very exciting nowadays to see a fM-level LOD under the presented testing configuration.

Response: We thank the reviewer for reviewing our manuscript and we appreciate the reviewer's constructive criticism. We would like to point out that the example paper referenced above (ACS Sens. 2017, 2, 2, 274) reports a Field-Effect-Transistor sensor, which is a well investigated device structure found in the literature as summarized in Table 1 and included in the references (Ref. 35, 44-47) of the manuscript. We would like to emphasize that the topic of our manuscript is a new device architecture; the first time a two-dimensional diode structure has been investigated for biomolecule sensing. However, in order to present the proof of concept, we have incorporated a well-known receptor-target (aptamer-cytokine) interaction. One key advantage of our sensing mechanism is the absence of a gate voltage, which does not require a third electrode needed in a FET sensor. The absence of a third electrode simplifies the fabrication process and required readout electronics. In order to clarify this point we have added the following content in the abstract of the main manuscript.

“To the best of our knowledge, this is the first time a two-dimensional (2D) diode is used for biomolecule sensing.”

“We have employed a well-known silane functionalization process and aptamer cytokine interaction to demonstrate the detection of cytokines using this new device architecture.”

“A key advantage of the diode sensor is that it uses only two electrodes, which simplifies the fabrication process and readout electronics compared to field-effect-transistors (FETs) that require a third gate electrode.”

Furthermore, there are key, noteworthy advantages of a diode sensor, as opposed to the extensively investigated FET sensors. In a FET sensor, gating is applied to bring the device to the linear regime of the transfer characteristic while exposed to an analyte solution; either by a liquid-gate or a solid-gate configuration. In a liquid-gated setup, a Ag/AgCl electrode is typically used which limits the portability and miniaturization of the device. Moreover, having an electric field applied via a liquid for a long time, can lead to undesirable leakage current to the electrodes that may result in electronic damage to the sensor. A solid-gate configuration (especially with a SiO₂ dielectric layer) requires higher gate voltages less desirable particularly for wearable applications [Nat. Phys. 4.5 (2008): 377-381]. In the two-electrode configuration of the diode sensor of this work, we avoid the issues arising from the presence of a gate electrode. We have added the following content to the introduction of the manuscript to emphasize the advantages of a diode sensor over a FET sensor.

“This two-electrode sensing mechanism is favoured over the common FET sensors, especially for portable and wearable applications. FET biosensors, including graphene-based FETs (GFETs), are generally used in a liquid-gated configuration with a Ag/AgCl electrode. This hinders the integration and scaling-down of the device [Biosensors and Bioelectronics 134, 16-23 (2019)]. The gate electric field, applied via a sample solution, can disturb the binding affinity between the charged cytokines and receptors, hence affecting the sensing stability [Physical review letters 95, 116104 (2005)]. Also, the continuous electrical stress in the liquid can lead to undesirable leakage current, that would generate false sensor response and electronically damage the sensor [ACS Sens. 2017, 2, 2, 274]. A serious issue related to TMD-based FET sensors is the hysteresis in the transfer characteristics arising due to the gate-modulated charges trapped at the TMD/dielectric interfaces. This behavior makes the current measured across the drain and source, under a given gate voltage, highly dependent on the sweep range, sweep direction, sweep time and loading history of gate voltage biases, which leads to inconsistent sensor readings [ACS Sens. 2017, 2, 2, 274]. A solid-gated FET employing a dielectric layer such as SiO₂ could mitigate some of the issues present in the liquid-gate sensors. However, the solid-gate FETs (specially with SiO₂ as the dielectric layer) typically requires high operating gate voltages in the range of 40 - 50 V for GFETs and even higher (100 V) for TMD-based FETs [ACS Sens. 2017, 2, 2, 274], which is unhealthy

for humans [Biosens. Bioelectron. 134, 16-23 (2019); Nat. Phys. 4, 377-381 (2008)]. By employing a simple diode-based mechanism, the issues associated with gating can be completely avoided.”

Reviewer comment: Second, the presented device fabrication process lacks a upscalable potential. I would like to recommend this manuscript transferred to other journals focusing on alternative biosensing techniques.

Response: We appreciate this comment from the reviewer. The manuscript demonstrated a proof-of-concept sensor based on geometrically asymmetric exfoliated MoS₂ flakes. However, this doesn't mean that the proposed device architecture is not scalable. In fact, large area graphene oxide films cut into asymmetric geometries have demonstrated diode behavior for measuring solution pH [Adv. Sci. 2, 1500062 (2015)]. Therefore, through patterning and etching of large area 2D materials, there is potential to fabricate large area arrays of asymmetric geometry sensors. There have also been advances in synthesizing wafer-scale 2D materials, such as single crystal graphene on germanium substrates [Science 344, 286-289 (2016)] and epitaxially grown MoS₂ on sapphire substrates [Nature Nanotechnology 16, 1201-1207 (2021)]. Asymmetric geometry sensors can therefore be scaled up in production using wafer-scale 2D material synthesis techniques. We have added the following content to the discussion section in the main manuscript.

“For commercial applications, a lithographic control over the flake geometry will be needed to make reproducible and reliable sensors for clinical usage with up-scalable potential. By employing a large-area compatible process to prepare thin films of MoS₂, the sensing area can be patterned into regular triangular shapes using lithography and etching to have better control over the flake geometry. With the advancement of large area 2D material synthesis, scale-up of diode sensors is possible. For example, large area graphene oxide films cut into asymmetric geometries have been employed to develop a nanofluidic pH sensor based on ion-current rectification [Adv. Sci. 2, 1500062 (2015)]. There have been advances in synthesizing wafer-scale 2D materials such as single crystal graphene on germanium substrates [Science 344, 286-289 (2016)] and epitaxially grown MoS₂ on sapphire substrates, which can be employed to scale up the sensor fabrication process [Nature Nanotechnology 16, 1201-1207 (2021)].”

Reviewer comment: Here are more comments: 1. The MoS₂ channel thickness is reported to be in the range of 13-60 nm. However, the optical micrograph inset in Fig. 3a shows a MoS₂ flake that looks significantly thicker than this range. Please use AFM to double check the thickness values.

Response: We appreciate this comment from the reviewer. In an effort to present an accurate picture of the device under test conditions, the inset of Fig. 3a showing an optical image was obtained while the sensing area was covered with a droplet of PBS. This made judging the flake thickness difficult and lead to confusion. We have modified the caption of Fig. 3a to be clear that the micrograph shows the sensor when coated in PBS.

“**Fig. 1 Performance of the sensor. a** An optical image of the measurement setup showing the sensing area 5 mm distant from the electrode pads for the probes. **The inset shows an optical micrograph of the device sensing area while coated in PBS.** **IV** response as a function of TNF- α cytokine concentration. **b** Log scale. **c** Magnified linear scale. **d** Normalized rectification factor, RF_N , as a function of TNF- α cytokine concentration. **e** Specificity of the CDS against two other non-targeted cytokines, IL-6 and C-reactive protein. **f** RF_N as a function of TNF- α cytokine concentration for a device with and without aptamers.”

Reviewer comment: 2. The authors claim that the rectification effect of the MoS₂ device is attributed to the different contact areas at two MoS₂/electrode interfaces. However, previous works have shown that many fabrication defects or contaminations can result in very strong rectification effects. How can these factors be ruled out?

Response: We appreciate this comment from the reviewer. Although fabrication defects could have some impact on the transport properties at the metal-semiconductor interface, the rectification behavior arising due to the asymmetric geometry of the 2D materials has been widely reported [Adv. Funct. Mater. 28, 1800657 (2018); Adv. Funct. Mater. 28, 1802954 (2018); Adv. Electron. Mater. 7, 2000964 (2021); Nanoscale 12, 7196-7205 (2020); ACS Appl. Mater. Interfaces. 13, 21499-21506 (2021)]. To further support that the rectification is due to the asymmetric geometry of the 2D flakes, we have added data from other symmetric and asymmetric devices fabricated under the same conditions to the supplementary file (Supplementary Fig. 3). The figures show that all devices fabricated with symmetric flakes did NOT exhibit noticeable rectification behavior whereas all asymmetric-based devices showed rectification effects. We have included the following content in the results section of the manuscript and have referred to the corresponding figure added to the supplementary file.

“The appearance of the rectification behavior due to the flake asymmetry is further supported by data from other asymmetric and symmetric MoS₂ devices fabricated in a similar method as shown in Supplementary Fig. 3. No significant rectification was observed for symmetric flakes.”

Supplementary Figure 2: Current-voltage curves from, **a, c, e**, three asymmetric devices with a noticeable rectification behavior. **b, d, f**, three symmetric devices with a very low or negligible rectification behavior.”

Reviewer comment: 3. It is still not clear why the diode-like I-V response characteristics result in the improvement of the biosensing sensitivity. A device physics model is needed to support the presented arguments.

Response: We thank the reviewer for this comment. We would like to emphasize that we did not claim that the diode-like-I-V response characteristics would result in an improvement of the biosensing activity, but rather that it offers a simplified two-terminal measurement setup for biomolecule detection based on the change in the rectification factor. We believe that the low detection level of the presented diode sensor at fM concentrations is achieved due to the device structure. With the two-step passivation method, we were able to have an Al₂O₃ thickness of 5 nm in the sensing area, which has enabled the detection of TNF- α concentrations as low as 10 fM. A previous work with an MoS₂ FET sensor employing a similar two-step passivation method has reported a LOD of 60 fM for TNF- α [Sci. Rep. 5, 1-13 (2015)]. However, the new device architecture of this work, employing a diode sensing mechanism, has enabled the detection of cytokines in a simpler manner compared to the common FET-based sensors due to the absence of a gate electrode. Furthermore, the aptamers used in this work have smaller sizes than other receptors (antibodies), making it possible to bring the charged cytokine molecules closer to the sensing area surface upon affinity binding, thus enhancing the sensitivity of the detection. This point has been mentioned in the discussion section of the manuscript as follows:

“We believe that the relatively low LOD was achieved due to the nature of the device structure consisting of a thin Al₂O₃ passivation layer (5 nm) over the sensing MoS₂ area, combined with a thicker passivation layer (75 nm) over the electrode areas which minimized leakage currents. A similar strategy was used for a MoS₂ FET sensor based on antibodies, employing a thick SiO₂ film to passivate the contacts and a thin HfO₂ layer on the MoS₂ channel, with a LOD of 60 fM. Since the aptamers used in this work are smaller in size compared to antibodies, they offer improved transportation of the target towards the sensor surface [*PROTEOMICS–Clinical Applications* 6, 563-573 (2012)]. Particularly the aptamer sequence used in this study, VR11, can bring the charged TNF- α closer to the sensor’s surface upon affinity binding [*Nanoscale* 10, 21681-21688 (2018)].”

Response to Reviewer #2

Reviewer comment: The paper reports detection of an inflammation biomarker using an asymmetric MoS₂ diode. Key noteworthy results are- small sample volume of 3 μ l, rapid measurement without the need for incubation and LOD of 10 fM.

Response: We thank the reviewer for their time in reviewing the manuscript and pointing out the key noteworthy results.

Reviewer comment: However several questions related to the detection mechanism along with the reproducibility and variability of the results need to be addressed.

1. The authors claim that the diode current increases due to increased hole current. To show enhanced ambipolar transport, please show the I_d vs V_G for negative V_D , negative V_G and positive V_D , positive V_G combinations before and after the various functionalization steps. One explanation for Fig. 4c data is an increase in negative charge in MoS₂ which opposes the claim of negative charge on the bio-molecules. Another explanation is increased gate leakage (current through Al₂O₃). Based on this, please show gate leakage data for these I-V measurements.

Response: We appreciate this comment from the reviewer as addressing it will help to clarify our detection mechanism. In this comment, the reviewer has raised two main points about the presented mechanism: (1) the enhanced ambipolar transport and (2) the trend in the I-V measurements presented in Fig. 4c and Fig. 4d.

Regarding the first point, we have included a typical transfer curve of our liquid gated MoS₂ devices with a full cycle scan of gate voltage V_{GS} in the supplementary document (Supplementary Fig. 8). In addition to commonly known n-type operation of MoS₂, additional hole current appeared under $V_{GS} < -0.25$ V which confirms the enhanced ambipolar transport present in our devices.

“

Supplementary Figure 3: Transfer curve (dual sweep) of the functionalized biosensor showing the ambipolar behavior. The hysteresis is related to the scanning speed of the gate bias which was 10 mV/sec.”

We have also added the following content to the results section in the manuscript.

“Similar to these observations, the increase in the current level with the cytokine concentrations/negative liquid gate voltage measured in cytokine diode sensor shows the existence of the ambipolar transport nature in the thin MoS₂ flakes used in this study which was confirmed by the transfer curve shown in Supplementary Fig. 8.”

Also, please find the additional transfer curves performed at positive and negative source-drain voltages.

Regarding the second concern about the increased gate leakage for the output curves shown in Fig. 4c, we have included the gate leakage data for the I-V curves shown in Fig. 4c (Supplementary Fig. 7b) that confirms that the gate current values were around 10^{-14} A for all the I-V sweeps. The following content was added to the results section in the main manuscript.

“Gate leakage current for each I-V response shown in Fig. 4c along with an AFM height image over the 5 nm Al_2O_3 film in the sensing area (showing the oxide layer is continuous and free of pinholes) is presented in Supplementary Fig. 7b and Fig. 7c, respectively.”

“

Supplementary Figure 4: a I-V response of the cytokine diode sensor (shown in Fig. 4c) in air. **b** Gate leakage current response for the drain-source current (I_{DS}) versus drain-source voltage (V_{DS}) for different applied gate voltages (V_{GS}) ranging from 0 V to -1 V shown in Fig. 4c. **c** AFM height image on the 5 nm Al_2O_3 film over the sensing area (in the trench).”

Finally, we would like to emphasize that the data in Fig. 4c and Fig. 4d were measured in absence of any biomolecules. We have presented the effect of the liquid gating on the diode I-V response to further confirm the biosensing mechanism. When negatively charged cytokines (in pH 7.4 sensing buffer) binds to the aptamer receptors on the surface of the 5 nm Al_2O_3 , it acts as a negative gate potential applied on the surface showing a similar trend to the negative gate voltage applied through the pure PBS droplet in Fig. 4c and Fig. 4d. Thus, these experimental results were incorporated to validate the claim that as the cytokines begin to bind, this induces a gating effect that changes the rectification factor of the diode sensor. We believe that any misunderstanding about having biomolecules in Fig. 4c might be due to a typo present in the caption of Fig. 4d, which has now been modified in the revised version of the manuscript. We have also added a label to the schematic in Fig. 4b to clarify that the measurement was done with a pure PBS droplet in absence of any cytokines. The modified figure and caption are shown below. We are sorry for the confusion that might have been caused due to this and we hope that the above response has clarified the mechanism and the data presented in Fig. 4.

“

Fig. 5 Detection mechanism. **a** Illustration of the folding of an aptamer, forming a compact G-quadruplex structure, due to the interaction of the TNF- α cytokine and aptamer, which brings negative charges closer to the Al_2O_3 thin film coated sensor surface. **b** Schematic diagram of a liquid gated asymmetric geometry diode sensor test setup to investigate the effect of gate voltage on IV response of the sensor. **c** Drain-source current (I_{DS}) versus drain-source voltage (V_{DS}) for different applied gate voltages (V_{GS}) ranging from 0 V to -1 V with an increment of -0.1 V steps (the gate voltage was applied via a pure PBS drop as shown in Fig. 4b). **d** RF_N versus V_{GS} of the liquid gated sensor.”

Reviewer comment: 2. Please provide AFM data of Al₂O₃ surface after the etch showing lack of pinholes and a continuous film.

Response: The reviewer makes a good point, and we realize that this information enhances the quality of the manuscript. We have performed an AFM measurement on the 5 nm Al₂O₃ surface covering the flake in the sensing area (Supplementary Fig. 7c). The AFM image confirms that the oxide layer in the sensing area (trench) is continuous and free of pinholes. We have included the following content in the main manuscript and referred to the respective figure in the supplementary document (Supplementary Fig. 7c).

“Gate leakage current for each I-V response shown in Fig. 4c along with an AFM height image over the 5 nm Al₂O₃ film in the sensing area (showing the oxide layer is continuous and free of pinholes) is presented in Supplementary Fig. 7b and Fig. 7c respectively.”

“

Supplementary Figure 6: a I-V response of the cytokine diode sensor (shown in Fig. 4c) in air. **b** Gate leakage current response for the drain-source current (I_{DS}) versus drain-source voltage (V_{DS}) for different applied gate voltages (V_{GS}) ranging from 0 V to -1 V shown in Fig. 4c. **c** AFM height image on the 5 nm Al_2O_3 film over the sensing area (in the trench).”

Reviewer comment: 3. Why is the data so noisy between -0.75 V and 0 V in Fig. 4c?

Response: We believe that the noisy behavior seen in Fig. 4c is due to the presence of PBS since the I-V response taken in air was steady as shown in supplementary Fig. 7a. For example, graphene FET sensors are known to be susceptible to similar fluctuations due to the disturbance occurring in the

capacitance across the electrical double layer (EDL) formed at the solution-graphene interface [Appl. Phys. Lett. 106.12 (2015): 123503]. Similarly, and since the Al_2O_3 thickness in the sensing area of the diode sensors is as low as 5 nm, we believe that the disturbance occurring in the electrical double layer during liquid-gating can lead to the fluctuations seen in Fig. 4c. We have added the following content in the results section of the manuscript and included the I-V response in air in the supplementary document (Supplementary Fig. 7a).

“The I-V response shows a noticeable fluctuation in current between -0.75 V and 0 V which we believe is due to the presence of PBS since the I-V response in air did not display such behavior (Supplementary Fig. 7a). It is reported that the graphene-based FET biosensors are susceptible to disturbance occurring in the capacitance across the electrical double layer formed at the solution-graphene interface [Biosens. Bioelectron. 134, 16-23 (2019); Appl. Phys. Lett. 106.12 (2015): 123503]. With the thin dielectric layer over the sensing area (5 nm), it is possible that our diode sensors undergo a similar disturbance during the liquid-gating which causes the current to fluctuate at low voltage values.”

“

Supplementary Figure 7: a I-V response of the cytokine diode sensor (shown in Fig. 4c) in air. **b** Gate leakage current response for the Drain-Source current (I_{DS}) versus drain-source voltage (V_{DS}) for different applied gate voltages (V_{GS}) ranging from 0 V to -1 V shown in Fig. 4c. **c** AFM height image on the 5 nm Al_2O_3 film over the sensing area (in the trench).”

Reviewer comment: 4. Were the devices washed before and after each of the functionalization steps? This would ensure covalent binding as reported and no physical adsorption.

Response: We thank the reviewer for this question, and we can confirm that the devices were washed between each functionalization step to make sure there is no physical absorption. This information is

included in the Methods section of the manuscript.

Reviewer comment: 5. What is max RF in a given device for a particular concentration? Is it $(\log|\max I_{+1V}| - \log|\max I_{-1V}|)$ for a single maximum value in a single graph?

Response: We thank the reviewer for the question, and we have realized that there is a typo in the definition of the RF. Please note that the terms need to be switched in the original formula as illustrated below. We apologize for the mistake, and it has been corrected in the revised manuscript.

“($\log|I_{-1V}| - \log|I_{+1V}|$)”

Due to the nature of the biomolecules, we can only do one set of experiments per device. In order to keep the conditions similar among different concentrations, the incubation time was strictly kept at two minutes. For each concentration, only one I-V response was taken after the two-minute incubation time. The figure of merit, RF, is calculated by subtracting the log of the absolute current at 1 V from the log of the absolute current at -1 V. The max RF is the maximum RF obtained in the data set for a particular device. It is typically obtained for the maximum concentration. The reason to define the RF max this way is to keep the normalized data between 0 and 1. However, we understand that the initial wording in the manuscript could lead to a bit of confusion and therefore we have modified it as follows:

“Due to variations from device-to-device in MoS₂ flake geometries and thicknesses, and in the aptamer functionalization process (fluctuations in room temperature and humidity), a normalized rectification factor, RF_N , was used to compare different devices, defined as:

$$RF_N = \frac{RF - RF_{PBS}}{RF_{max} - RF_{PBS}},$$

where RF_{PBS} is the RF with only PBS (0 fM TNF- α in **Fig. 4b** and **Fig. 4c**) and RF_{max} is the maximum RF for each individual device (which is typically at the maximum concentration).”

Reviewer comment: 6. Please provide data from various devices to show reproducibility and variability with error bars.

Response: In order to confirm the repeatability, each experiment has been repeated few times. However, due to the nature of the experiments which involves biological samples, we could only repeat one set of experiments, one time for each sensor [ACS Nano 2021, 15 (7), 11461-11469]. Since the main purpose of this work is to present the novel detection mechanism, the devices were

fabricated using exfoliated MoS₂ flakes. A major disadvantage of using exfoliated flakes is that we do not have a control over the dimension and the shape of the flakes. Therefore, each device possessed a different initial RF, resulting in varying response from device to device. Nevertheless, we were able to confirm the overall trend of the cytokine sensor by repeating the test on several devices where, with the increase of the cytokine concentration, an increase in the figure of merit (RF_N) was observed. We have included the data from five different devices in the supplementary document (Supplementary Fig. 10) as shown below.

However, due to the difference in the rectification at the pristine stage, the linear ranges and the saturation points were different from device to device. Therefore, we were not able to generalize the data set for multiple devices which prevented us from calculating the mean trend line and error bars.

A similar observation has reported for a WSe₂ COVID sensor, fabricated using CVD grown WSe₂ flakes [ACS Nano 2021, 15 (7), 11461-11469]. Due to the difference in the distribution and the number of flakes on each device, the starting current levels and the change of the figure of merit with the increase of target concentrations were different from device to device, which was illustrated in their supplementary document. Due to this reason, the standard deviation calculations are not valid since the responses were not at the same scale.

“

Supplementary Figure 8: Variation of the normalized RF with increase of the TNF- α concentration. The data has been normalized so that the response in pure PBS was set to zero. Response for a diode sensor fabricated with a the two-step Al_2O_3 deposition method where the Al_2O_3 thickness over the sensing area is 5 nm. b same method as a. c 15 nm thick one-step Al_2O_3 deposition method. This device was only tested for the 4 cytokine concentrations (starting at 0.001 nM) as shown in the graph but still shows the same trend. d 20 nm thick one-step Al_2O_3 deposition method. This device was mainly tested for lower concentrations starting from 0.0001

nM to 0.01nM which shows the same trend. e 40 nm thick one-step Al₂O₃ deposition method, showing the general trend.”

Reviewer comment: 7. Is there a way to confirm the increase in negative charge on the biomolecules after cytokine binding through say KPFM or other measurement techniques? Also, for the bending of the molecules?

Response: We thank the reviewer for this comment. We have included a fluorescence measurement (Supplementary Fig. 6) that shows that the fluorescence of the FAM dye attached to the aptamers is dimmed upon introducing cytokines. The aptamers that are covalently bonded to the surface are unfolded with the FAM dye end further away from the surface. Upon addition of cytokines, the aptamer folds to a new conformation bringing the FAM dye closer to the surface (Fig. 4a) which causes the weakening of the fluorescence signal. The following content has been also added to the results section of the manuscript.

“Since the aptamer changes its form upon binding to a target, the fluorescence intensity changes depend on the manner of the FAM dye modification. In this case, the fluorescence intensity is expected to decrease after the cytokine interaction due to the aptamer folding. Hence, the decrease in the fluorescence intensity could be regarded as an indicator of the change in the aptamer structure into a compact form, bringing the charged cytokine closer to the surface as shown in Supplementary Fig. 6.”

“

Supplementary Figure 9: Fluorescence spectrum of the aptamer functionalized surface before and after the interaction with TNF- α cytokines at a concentration of 58.5 nM.”

REVIEWERS' COMMENTS

Reviewer #3 (Remarks to the Author):

I reviewed the manuscript and mainly evaluated the authors' responses to Reviewer 2's questions and comments. I found the authors' responses adequate and clear.